# Creating a Virtuous Food Cycle in Monterrey, Mexico

**Rob Roggema** [1,*], **Ana Elena Mallet** [2] and **Aleksandra Krstikj** [2]

1 Tecnologico de Monterrey, Escuela de Arquitectura, Arte y Diseño, Campus Monterrey, Monterrey 64849, Mexico

2 Tecnológico de Monterrey, School of Architecture, Art and Design, Mexico City 04510, Mexico; aemallet@tec.mx (A.E.M.); sandra.krstik@tec.mx (A.K.)

* Correspondence: rob.roggema@tec.mx

**Abstract:** This article focuses on the connection between health, food, and social wellbeing. Several health problems are caused by the types of food consumed. At the same time, traditional ingredients are increasingly less used. The potential of healthy, traditional foods, locally grown ingredients, and preparation in local quality restaurants could decrease health problems, especially in disadvantaged communities. The methodology focuses on developing the missing links between traditional recipes, the growth of local food, and the preparation and consumption of healthy food. The main finding of this article is the interconnected model and the definition of the components that link the abovementioned separate fields. Major components are a collection of traditional recipes from the elderly in local communities, establishing an urban farm in disadvantaged communities, local quality chefs who use the recipes and ingredients in their menu, and pop-up restaurants in the local communities where residents have access to healthy dishes. The main conclusion is that by connecting the fragmented fields of investigation, the most vulnerable residents gain access to healthy food options.

**Keywords:** traditional recipes; urban agriculture; disadvantaged community; healthy food; mental and physical health; restaurant chefs

## 1. Introduction

To provide a healthy diet for everyone and keep the planet within its planetary boundaries [1], a fundamental shift in the way food is produced, processed, and consumed is needed. Food systems have three components: food supply chains, food environments, and consumer behavior. While the food supply chain refers to the processes that describe how food from a farm reaches consumers (production, processing, distribution, consumption, and disposal of food), food environment refers to the opportunities, environments, and physical, economic, political, and socio-cultural conditions that frame the interaction of people with the food system and shape decisions about food acquisition and consumption [2]. Several problems are associated with the current food system. Additives used in processed foods, such as synthetic antioxidants or nitrite in sausages may reduce people's physical and mental health due to the formation of lipids, oxidation of protein radicals, or the additives themselves. Food security is under serious threat from climate change. Lastly, the fresh vegetables and ingredients used in traditional recipes generally have a positive impact on human health, although excessive use of oil and salt can be a health risk. In any case, when these recipes are on the brink of becoming extinct, witholds them from playing their potential role in the transformation towards a healthier diet.

For each of these aspects, research has been undertaken, and solutions have been proposed. However, most of the research has produced stand-alone outcomes that are separated from other aspects. In this article, this fragmented food field is analyzed, and suggestions will be made to close the gaps in between. Therefore, a proposal is presented on how the cycle of a healthy food system can be closed, taking the city of Monterrey in Mexico as an example.

## 2. Problems Related to the Food System

The following problems in the current food system are seen as crucial flaws for a healthy diet: physical and mental health, climate change, and loss of food heritage. Other aspects of the food system, such as economics, pandemics, or environmental impacts, are also problematic but are excluded from this article because these symptoms are only indirectly related to actionable interventions in the local/regional food system.

### 2.1. Physical Health

In Mexico, obesity is a serious problem that is accelerating, partly caused by the influence of the food industry on national health policies [3]. The increase in obesity is linked to the consumption of added fats, sugars, and refined grains and of the snacks, sweets, beverages, and fast foods in which they are prominent [4], all elements of the food system the food industry has an interest in. The change in caloric quantity and quality of the food supply combined with a food industry that produced and marketed convenient, highly processed foods from cheap agricultural inputs, result in higher numbers of obese people. The high amounts of salt, sugar, fat, and flavor additives in these industrialized foods and the way these foods are manufactured as extremely appetitive meals have driven overconsumption. Moreover, the easy and ubiquitous access to these foods, often convenient and inexpensive, has transformed the eating behavior of many, with people snacking and eating in restaurants more while cooking fewer meals at home [5]. The national rate of overweight adults has risen significantly in the last two decades, especially in the northern region where the city of Monterrey is the largest urban area [6]. It has been recorded that people from the northern region have a higher prevalence of obesity (41.6%) than people from the central (33%) and southern regions (36.1%) [6] (p. 685). Additionally, studies on food group consumption in the country revealed that belonging to urban localities, northern regions, and middle and high terciles of the wellbeing index was associated with higher consumption of eggs and dairy products, as well as higher consumption of processed meats, snacks, sweets, and desserts [7]. Thus, obesity related to processed food overconsumption is currently considered one of the major public health problems in the country, particularly in urban areas of the north.

Children are particularly increasingly obese and overweight. The presence of overweight and obesity in children and adolescents in Mexico is between 35 and 40%, respectively; thus, it is considered a tragic problem that has placed Mexico as the country with the highest prevalence of these conditions in the world [8]. In the northern region, a high percentage of depression among adolescents with overweight and obesity was observed (57.9%) [8]. The epidemic of obesity in children is explained by the lower amounts of fats/oils, vegetables/soups, breads/grains, mixed meats, desserts, candy, and eggs, and increasing amounts of fruits/fruit juices, beverages, poultry, snacks, condiments, and cheese they take in. In parallel, changes in eating patterns, such as the number of meals eaten at restaurants, food availability, portion sizes, snacking, and meal-skipping, explain the increase in severe or even morbid overweight among children. For example, in the northern regions of Mexico, the preschool population was the largest consumer group of processed meats not recommended for daily consumption (36% consumed this product at least three days/week) [7].

Apart from obesity, the risk of diabetes is also becoming problematic. Increased fast food consumption implies a higher risk of incident gestational diabetes [9], while the consumption of Ultra Processed Food is associated with an increased risk of type II diabetes mellitus [10]. The risk of type II diabetes can be reduced with higher intakes of green vegetables, fruit and berries, oil and margarine, and poultry [11].

The danger of an industrial diet [12] is the overconsumption of salt, sugar, and fats. The food giants have made people addicted to the intake of these unhealthy food products [13–15]. Ultra-processed foods are thought to be related to many chronic diseases, such as inflammatory bowel diseases and metabolic syndrome [16]. The neoliberal diet brings about healthy profits for the food industry but makes people unhealthy [17].

## 2.2. Mental Health

Not only does the current food intake impact people's physical health, but it also has profound consequences for mental health. Stress and depression are deemed related to food insecurity [18] and the consumption of unhealthy foods, such as sweets, cookies, snacks, and fast food. On the contrary, the consumption of healthy food, such as fresh fruits, salads, and cooked vegetables, reduces perceived stress and depressive symptoms [19].

A range of mental health implications arises from the type of food consumed [20]. As an example, depression and anxiety are lower, people are better at coping with negative conditions, and the overall quality of life is higher amongst people consuming a Mediterranean diet, which contains relatively higher amounts of vegetables, fruits, legumes, nuts, beans, cereals, grains, fish, and unsaturated fats such as olive oil. Moreover, people who consume more vegetables experience less stress and more happiness. Positive effects on a range of mental health aspects are found for the consumption of fruit, nuts, legumes and a greater diversity of vegetables, and fruits, and reducing the intake of takeaway food and snacks (Table 1).

**Table 1.** Overview of relation between type of food and mental health (adapted from [20]).

| Type of Food/Diet | Relation to Mental Health Aspects |
| --- | --- |
| Mediterranean diet | Lower depression and anxiety<br>Negative affect<br>Better coping<br>Overall quality of life |
| Higher vegetable consumption | Less stress<br>More positive emotions and happiness |
| Higher fruit consumption | Less anxiety<br>More positive emotions and relationships. |
| Higher intake of nuts | Reduced depression, anxiety, and stress<br>Better mental health, self-worth, and overall quality of life |
| More legumes | Reduced anxiety, stress, and negative emotions<br>Greater coping<br>Psychosocial score<br>Overall quality of life |
| Greater diversity of vegetables | Reduced depression, anxiety, and negative emotions<br>Higher positive emotions |
| Greater diversity of vegetables and fruits | Higher independent living,<br>Improved mental health,<br>Higher happiness<br>Better relationships<br>Psychosocial score<br>Overall quality of life |
| Reduced intake of takeaway food | Better pain<br>Overall physical health<br>Improved mental health |
| Reduced intake of unhealthy snacks | Coping<br>Psychosocial scores |

## 2.3. Climate Change

Growing food in Mexico is under pressure due to rapid climate change. Temperatures are expected to rise more than two degrees Celsius by 2050 [21], and precipitation is expected to decrease in nearly all future scenarios by maximally 5.6% in the warmest season. This indirectly leads to a 34% increase in agricultural droughts and a 29% increase in hydrological droughts, which are expected to occur 74% more frequently. Moreover, the duration of heat waves is expected to increase by 4.005%, occurring 98% more often, while groundwater recharge is reduced by nearly 5% [21]. Almost twice as many people will be affected by river flooding, and the risk of water stress is deemed extremely high. At the same time, the water demand in the agricultural sector is expected to rise by nearly 19%. Because of this complex of reasons, it is suggested that climate change could drastically reduce Mexico's agricultural productivity leading to serious socio-ecological impacts [22].

Rural and peri-urban areas, especially in low-and middle-income countries, are the main source of food growth, consumption by its residents, and income [23]. It is highly

sensitive to temperature and precipitation changes [24]. Vulnerable societies are the most affected by climate change impacting food production, access, and price. It impacts not only local communities [25] but has significant knock-on effects at the regional and even global scale [26].

Under a high-emission scenario, large reductions in yields are expected in Mexico by the end of this century. A loss of productivity for maize (up to 42%), rice (up to 51.4%), sorghum (up to 41.1%), soybean (up to 59.1%), wheat (up to 23.3%), and sugarcane (up to 11.7%) represents an economic reduction of value of 37,934 million dollars nationally.

In the state of Nuevo Leon (NL), yields for maize and sorghum are expected to drop by 50–70% in 2061–2099 relative to 1980–2012 as follows: up to 30% for wheat and over 70% for soybeans [27].

Amongst several other regions in Mexico, the problems in the north(east) regarding temperature, precipitation, and population vulnerability (Figure 1) are serious. This implies that it is urgent to adopt more sustainable use of the land in Mexico that also provides adequate nutrition for the population by 2050. In an integrated way, this could be achieved by simultaneously limiting agricultural expansion, for instance, by more localized and nutritious foodscapes, reducing emissions, and expanding forested lands. The plans for climate biodiversity and agriculture at the national level should therefore include the transformation to more sustainable diets [28], consisting of lower intake of animal-source foods, unhealthy foods (refined grains, added sugar and fats, mixed processed dishes, and sweetened beverages), and higher intake of fruits, vegetables, and whole grains. This diet pattern is already common in disadvantaged populations. Environmental sustainability can be achieved when the nutritional quality of diets is enhanced through an increase in the consumption of legumes, fruits, and vegetables and a reduction in unhealthy foods [29].

*2.4. Loss of Food Heritage*

A strong attachment to traditional food systems improves people's diet quality [30]. In combination with the existing food patterns in vulnerable communities, it may be problematic that food patterns of more traditional heritage are on the brink of becoming extinct. The knowledge and understanding of traditional recipes and the use of ingredients are under threat due to the rapid transition of younger people to industrialized food patterns. For example, a recent qualitative study on the process of colonization of food practices of migrant indigenous women residing in the Monterrey metropolitan area found that when the participants lived in rural areas, they mainly maintained their traditional indigenous diet, and their food came directly from their harvests, backyards, and local sales. However, when indigenous women migrated to the city, their diet changed to a "mestizo" diet, which is based on the consumption of both industrialized products and traditional foods [31]. Roman and Gaspar (2022) state, "A characteristic of this type of diet is that it has become a solitary act in which individuals are totally disconnected from food production and culinary preparation . . . . the time dedicated to eating has decreased, people eat at irregular times, family meals have become less frequent and at the same time, the consumption of fast or street food has increased . . . This contrasts with the traditional indigenous diet, where most or all the foods consumed are known, that is, there is information about where it comes from and what is contained in what is being eaten." [31].

Grandmothers signal, for instance, that the "tiendas" easy-to-prepare foods have replaced ancestral ways of food preparation that practice grinding, peeling, and log cooking. Moreover, young mothers dismiss the knowledge held by their grandmothers, and this stands in the way of retaining traditional meals [32]. With an aging population, this is a real risk. This implies an eventual reduction in nutritional food quality for the population in general. It is necessary to save these endangered foods for our future food security, the good of the planet, and the good of our own health. These are precious resources that were a long time in the making [33]. Phenolic compounds such as rice, beans, enchiladas, tomatoes, lettuce, and corn are found in many traditional Mexican meals and have a positive effect on

health conditions [34]. Without awareness of the positive health implications of traditional food patterns, their availability will rapidly reduce and limit access to healthy food.

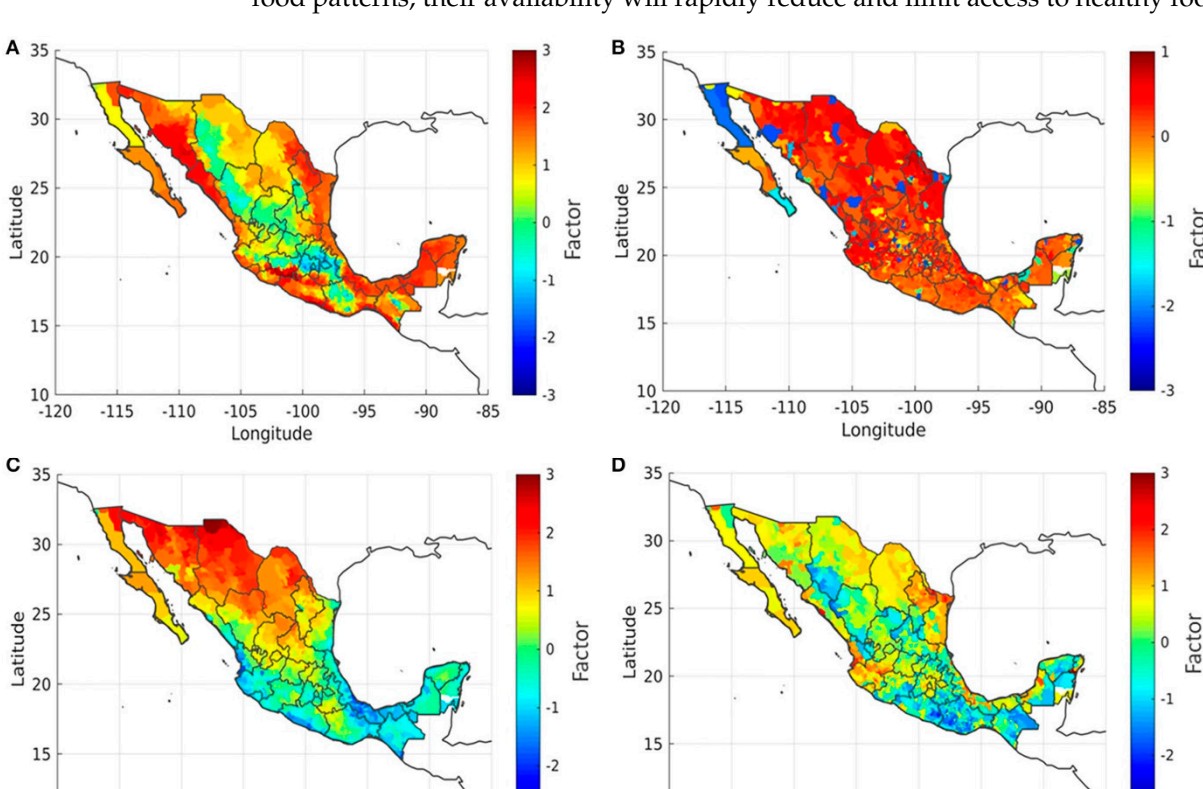

**Figure 1.** Climatic and socio-economic factors regionalization. (**A**) Combination of variables related to temperature, (**B**) combination of population and GDP size, (**C**) combination of temperature variables, precipitation, and rainfed maize yield, and (**D**) combination of index of marginalization, people working on the primary sector and per capita GDP [22].

UNESCO designated 2010 traditional Mexican cuisine as a cultural model that includes agricultural activities, ritual practices, ancient practical knowledge, culinary techniques, and ancestral community behaviors. The Mexican culinary tradition has been enriched through cultural contacts since pre-colonial times, where the different ethnic groups that inhabited what we now know as Mexico exchanged products such as cocoa, corn, vanilla, etc. [35]. The traditional food of Nuevo Leon is a mixture of the cuisine of the indigenous population and Spanish Catholic and Jewish cuisine. One important protagonist in this cuisine is meat, and the drying technique for food preservation is common due to the extremely hot climate and infertile lands. From grains, the most common ones produced are maize, wheat, and potatoes; from legumes are beans; from vegetables are tomatoes, nopales, and squash; from fruits are mandarins, oranges, lemons, and bananas. Eggs and dairy products are also frequent ingredients in traditional meals; thus, a famous local dish is "machacado con huevo"—a dish prepared with beef jerky, eggs, and chili sauce, often accompanied by fried beans and maize or wheat tortillas. Another widely known traditional food is "tamales"—(from tamalli, that means wrapped), a dish originating in the northern Mexican highlands that was already prepared before the arrival of the Spanish; the preparation consists of a corn dough to which some type of filling is added that can be salty (meat, stew, chicken) or sweet (fruits); then, it is wrapped in corn or banana leaves and cooked in steam [35]. Figure 2 presents a graphical description of the inclusive community process of preparing tamales.

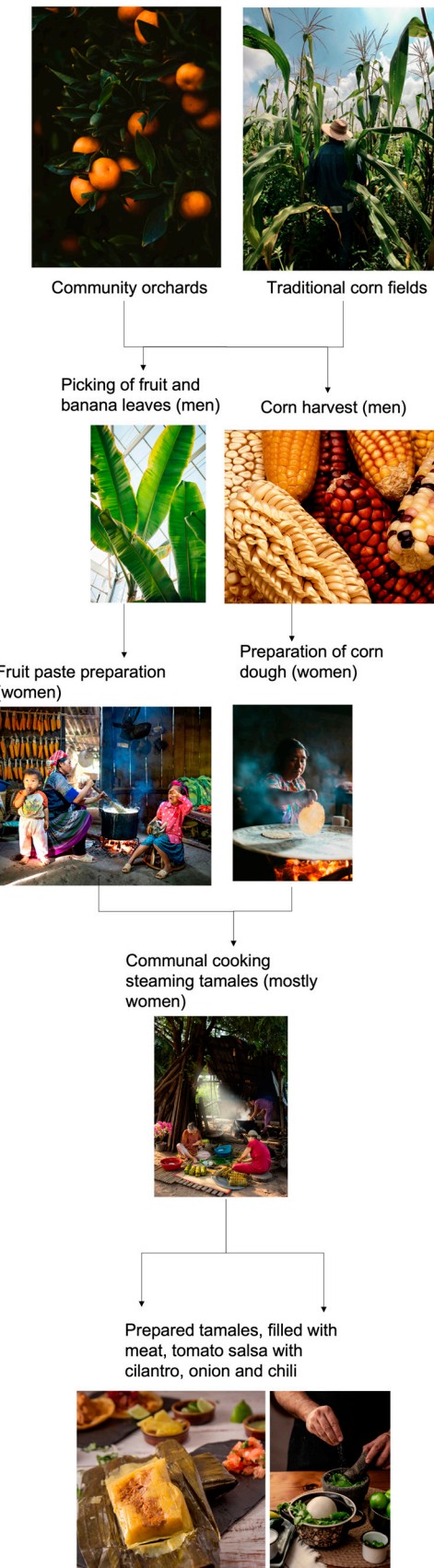

**Figure 2.** Graphical abstract of the process of preparing tamales (free images used from the platforms Unsplash, Pixabay, and Pexels).

*2.5. Fragmentation*

In conclusion, the food provision of local communities is impacted by an integrated complexity of problems, including physical and mental health, climate change, and the loss of nutritious food recipes and ingredients. It leads to the degradation of food quality and reduces population health. For each of these problems, field policies and solutions are proposed, and in-depth research is carried out. However, an integrated approach that effectively improves people's health is hindered by the separated building blocks that each solve part of the problem:

- There are many initiatives for growing food in the city. Urban agriculture gardens, balconies, food forests, and many other forms are proposed and realized;
- Traditional recipes are collected in books and preserved for the broader (elite) public;
- Programs for improving the quality of life in vulnerable communities, social justice, and equity are extensively undertaken;
- Many innovative restaurant chefs want to use local ingredients to create innovative dishes;
- Climate change mitigation and adaptation is embedded in global, national, and regional plans and programs;

The main issue is not that solutions are not conceived but that they are implemented apart from each other. The gaps between climate policy, urban agriculture projects, disadvantaged communities, heritage dishes, and restaurants and consumers are standing in the way of ultimately improving people's health through nutritious and sustainable food provision.

**3. Methodology**

In response to the problem of disintegrated solutions, the methodology of this work consists of the development of a model to reintegrate separate solutions (Figure 3).

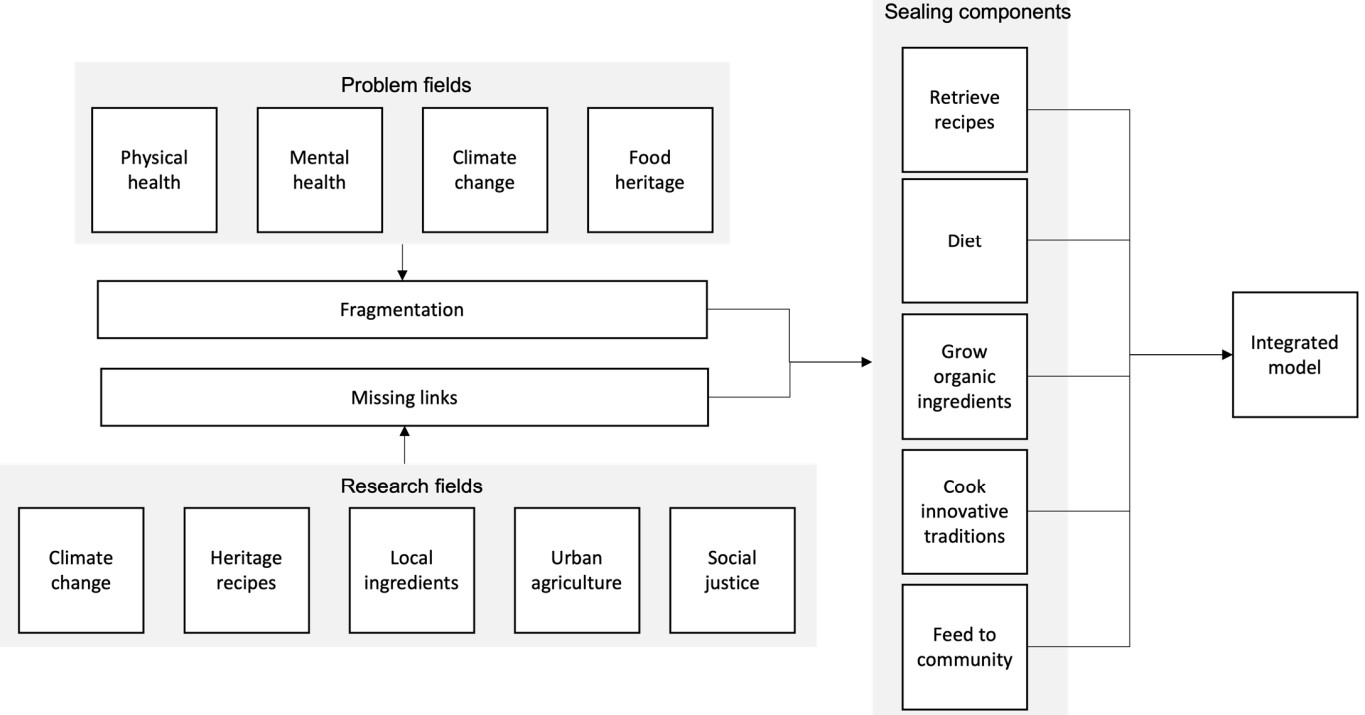

**Figure 3.** Diagram of the methodology.

The first part of the research consists of an analysis of current problem fields related to the food growing system. From the literature, the current insights into the relationship between the food system and physical health, mental health, climate change, and the

heritage of food, respectively, are critically investigated. From this, the existing coherence, or absence thereof, is illuminated. In the second part, current research fields are explored. For each of the fields, the state of the art has been investigated, and eventual missing links between them are identified. The two parts are subsequently brought together to define possible so-called sealing components, elements that can bridge the gaps between the current research fields and collectively solve problems found related to the food system. Each of the sealing components has been preliminary explored to test their realism. This part of the research is undertaken through a series of semi-structured interviews (approx. 10) with local community leaders, restaurant chefs, and urban farmers. The final component of the research is to converge the elements of the separate research fields together with the sealing components into one logic model.

## 4. Results

### 4.1. Outline of the Model

Like in most cases, the model (Figure 4) places the distinct fields of investigation (the building blocks) outside the core, feeding their knowledge and solutions to the coherent inner circle. This internal circle consists of connected pieces that determine the integrated whole. No single component can be extracted, as this would mean the virtuous circle will be broken. Finally, the model presupposes a continuum, from finding traditional recipes to eating healthy dishes. This will rekindle the collection of more recipes and so on. The circle gains strength when it is repeatedly closed.

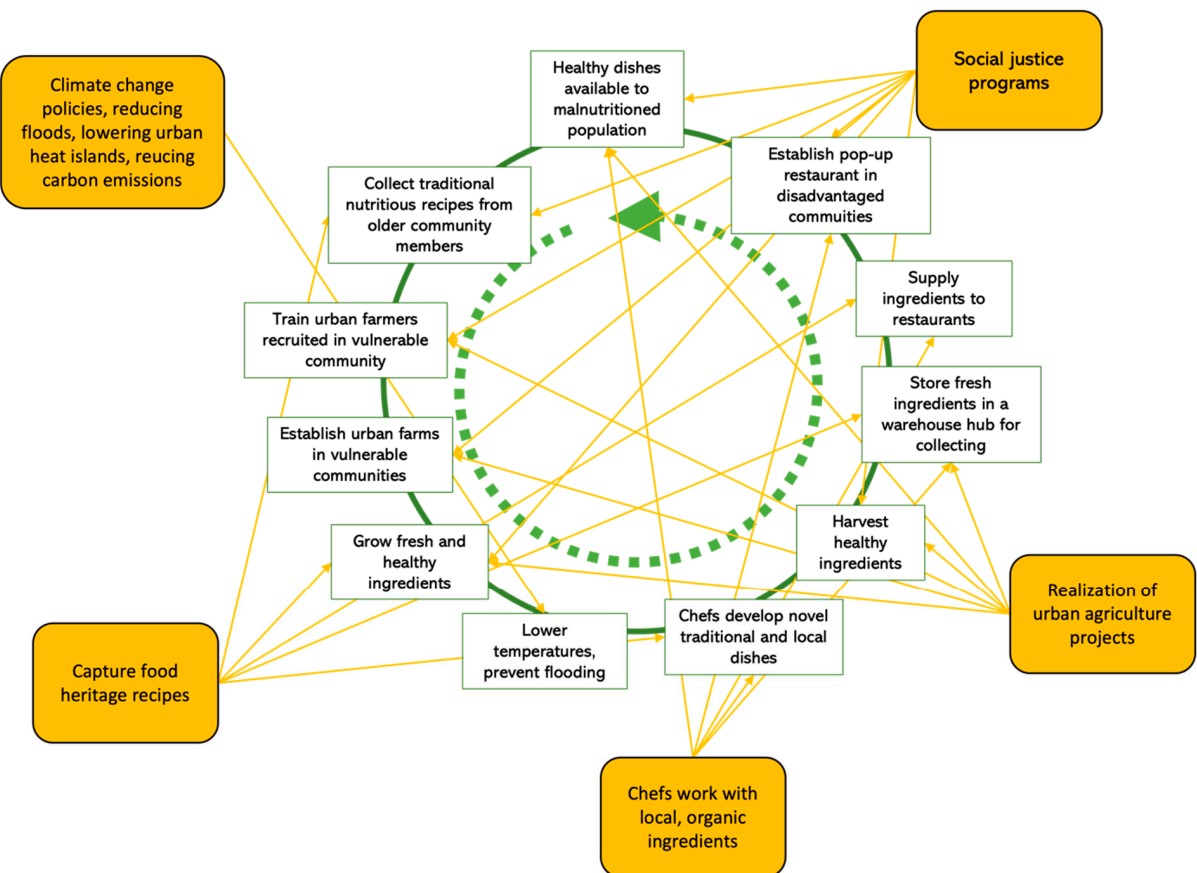

**Figure 4.** Model of the virtuous circle.

### 4.2. Building Blocks of Existing Knowledge

The field of science is currently divided into a multitude of specialistic parts, each delivering the outcomes that fit the scope of their specialization. Out of this research, valuable results are derived; however, for more holistic problems, this may sometimes

work counterproductively. The research fields that are connected to the food system often have connections with other topics as well. However, the outcomes that bring new insights in one of the research fields may compromise solutions that are urgent in another. In this section, the current status of several of these fields is discussed.

### 4.2.1. Climate Change Policy

For many years, science has been clear about the changes in climate: rising temperatures, caused by increasing GHH emissions, influenced by man, leading to a range of impacts that threaten human livability on earth, such as droughts, flooding, heat, bushfires, and rising sea levels [36]. The Paris Agreement [37] has indicated that global warming should be limited to a maximum of 1.5 degree Celsius; however, latest findings show that we are on track to two degrees or even more, and there is 'no credible pathway to 1.5 C limit' [38]. Every city, region, and country works on climate change mitigation and adaptation plans. Concretely, the City of Monterrey translates the Mexican national climate policy [39] and the new climate change law [40,41] in policies for climate change mitigation and adaptation [42], specifically the impact of heat [43], and develops a resilient city strategy [44,45]. The focus lies on adapting to heat, prolonged droughts, and occasional severe flooding.

Decision support tools and approaches are clearly improving rapidly; however, factors such as resistance to change, the cautious approach by development agencies, governance structure and overlapping jurisdictions, funding, and limited community engagement remain, in many cases, obstacles to successful implementation of ecosystem-based solutions [46]. For example, a recent project performed to improve climate adaptation by the Instituto de la Vivienda de Nuevo Leon (INFONAVIT) was ViDA: a low-carbon social housing project, which would introduce a series of design principles to conventional social housing that would significantly reduce the energy resources used by the house during its design life and therefore considerably improve the quality of life provided to residents. The project ViDA stood for low-income people and protected the notion that they had the same rights to the benefits provided by climate change action as the elites who could afford to upgrade their homes without assistance. Nevertheless, the outcomes of the project point to the difficulties the poor face in claiming these rights, with some examples as follows: (1) residents have not been able to claim the long-term savings promised by the dwellings; (2) residents neglected open spaces built into the planning design of the development; (3) residents have constructed walls to barricade their homes in response to the need for security in response to the 'drug wars' which came into suburban Monterrey; (4) residents have constructed additional living and working spaces in response to their dependence on home-based small enterprise for their livelihoods [47].

The actions that follow from climate change research and land in local and regional policies focus on the reduction of greenhouse gas emissions and the adaptation to impacts of climate change, e.g., dealing with floods, water shortages, and urban heat. Though these efforts undoubtedly are beneficial to establish social and livable neighborhoods and improve food growing conditions, etc., a direct relation is absent. Climate policies, generally, do not aim to improve social safety or stimulate urban farming.

### 4.2.2. Social Justice Programs

With many people living in vulnerable and poor conditions and half of the global population living on less than USD 6.85 per person per day [48], social justice and equity are of global concern [49]. SDG-10 is explicitly devoted to reducing inequality within and between countries [50]. Still, "over one billion people in the world live in unacceptable conditions of poverty, mostly in developing countries. Poverty manifests itself in the form of hunger and malnutrition, ill health, lack of access to education and basic services, increased morbidity and mortality from illness, homelessness and inadequate housing, an unsafe environment, and social discrimination and exclusion" [51]. The lack of urban planning and risk management makes the problems of habitability more pronounced,

restricting the possibilities of aging in the place and the capacity to adapt to the challenges of climate change. For example, in cities such as Monterrey, studies reveal the high social vulnerability of older adults to flooding and their limited capacity to adapt to natural hazards associated with climate change [52]. To improve this situation, it is suggested to (1) make farming more rewarding, (2) train skills income generation by young people, (3) offer land grants for young people, and (4) stimulate cooperation between urban and rural youth in food production and distribution [51]. From this general objective, it is a long way to implementation. At the same time, bottom-up initiatives to enhance the quality of life of vulnerable communities are undertaken. In la Campana, City of Monterrey, Mexico, a long-lasting program is executed for which an extensive social analysis is presented [53], propositions to upgrade the settlement through urban governance [54], and initial steps are undertaken to improve the quality of the river Arroyo [55,56], the public space for children [57], and establishing pocket parks in the area [58].

Both approaches, global top-down policies and local bottom-up projects, are necessary. However, it is deemed difficult to connect the two in a coherent implementation in synergy with other objectives. Though social programs are implemented in disadvantaged neighborhoods, they generally do not enlarge opportunities for establishing local restaurants, growing organic food, or reducing the impact of flooding.

### 4.2.3. Food Heritage Recipes

In recent years, the collection of traditional recipes has taken flight. Existing heritage recipes are collected in different parts of the world: Alaska [59], England [60], Hungary [61], and Nuevo León [62]. Most of these collections focus on providing recipes for (privileged) people who want to cook at home. A direct relationship with nutritional value, the impact of the food system on the broader planetary health, or disadvantaged communities is absent.

Apart from the traditional recipes, which are collected for Mexico and Nuevo León also, traditional Mexican menus are analyzed. This research emphasizes the food items that have been used most in the Mexican kitchen [63] and is carried out for Mexico as a whole (Table 2) as well as other regions, such as northern Mexico (Table 3). It gives the categories, but not the amounts of food to be taken for a healthy diet.

**Table 2.** Most cited items and food groups refer to all regions of Mexico [63].

| a. Food Groups Present in at Least 50% of the Studies | | | | | | | | | |
|---|---|---|---|---|---|---|---|---|---|
| Grains and Tubers | Maize Products | Legumes | Vegetables | Fruits | Oils and Fats | Beverages | Meats | Sweet and Sweeteners | Herbs and Condiments |
| Maize, amaranth, rice, wheat (as bread, pasta, tortillas), potato, sweet potato, yucca | Tortillas, tamales, atole, soups (pozole, menudo), other | Beans | Squash, chayote, nopales, tomato, tomatillo, carrot, lettuce, purslane, quelites, quintoniles, mushrooms, huitlacoche, squash blossoms | Anona, apple, banana, berries, capulin, citrus fruits, guava, guanabana, jicama, mamey, mango, melon, papaya, peach, pear, pineapple, pitahaya, plums, tejocote, prickly pear, zapote | Avocado, vegetable oil, cream | Chocolate drinks, pulque, tesgüino, coffee, aguas frescas, natural fruit juice | Turkey, chicken, venison, pork, rabbit, beef, lamb, chevon, dogs | Honey, pan dulce, sugar and sugarcane, desserts, sweets | Annato, acuyo, chile, coriander, epazote, garlic, onion, parsley, pepper, vanilla |

**Table 2.** *Cont.*

| b. | Food groups present in at least 50% of the studies | | | |
|---|---|---|---|---|
| **Nuts and seeds** | **Fish and seafood** | **Dairy** | **Eggs** | **Insects** |
| Peanuts, pumpkin seeds, chia seeds, sesame seeds | Shrimp | Cheese, milk | Chicken eggs | Grasshoppers and locusts, ants and their larvae |

**Table 3.** Food groups of Northern Mexico [63].

| a. | Food Groups Present in at Least 75% of the Studies | | | |
|---|---|---|---|---|
| **Grains and Tubers** | **Legumes** | **Vegetable** | **Fruits** | **Eggs** |
| Maize, amaranth, rice, wheat (as bread, pasta, tortillas), potatoes | Beans | Squash, tomato, nopales, guaje, quelites, mushrooms | Banana, citrus, fruits, prickly pear | - |

| b. | Food groups present in at least 50% of the studies | | | | |
|---|---|---|---|---|---|
| **Maize products** | **Beverages** | **Fish and seafood** | **Meats** | **Sweets and sweeteners** | **Herbs and condiments** |
| Tortillas, pinole | Beer, coffee, soda | - | Chicken | Sugar and sugarcane | Chile, onion |

These food groups can be interlinked with the proposed Healthy Reference Diet to support human and environmental health [1]. The EAT-Lancet Commission [64] found that the following should be consumed to achieve this goal: moderate consumption of whole grains, starchy vegetables, vegetables, fruits, and legumes; moderate or low amounts of seafood and poultry; low or no amounts of red meat, refined grains, and added sugars [65]. In energy terms, this implies a feasible average daily intake of 2288 kcal per capita (780 kcal for cereals, 236 kcal for fruits and veggies, 228 kcal for oils and fats, 148 kcal for sugar, 22 kcal for fish, 22 kcal for pork, 60 kcal for eggs, 68 kcal for red meat, 96 kcal for poultry, 149 kcal for milk, 240 kcal for pulses, 52 kcal for roots, 82 kcal for nuts, 11 kcal for beverages and spices, 53 kcal for alcohol, 13 kcal for offal ("other"), and 28 kcal for animal fat [66]). This diet is adapted to the Mexican context [61] showing the average amounts of intake per category in grams/day [65] and what this is equivalent to in terms of size of volume (Table 4).

**Table 4.** Average intake in grams per day per food group.

| Food Group | Gram/Day | Equivalent to |
|---|---|---|
| HF Cereals | 125–232 | 5–9 cups |
| Tubers | 50–100 | About 0.1 Celeriac |
| Vegetables | 200–600 | 2²/₃–8 cups |
| Fruits | 100–300 | 1–2 apples |
| Dairy | 250–500 | 1–2 cups |
| Red meat | 14–28 | Size of a stack of 7–14 playing cards |
| Poultry | 29–58 | Size of a stack of 14–28 playing cards |
| Eggs | 13–40 | ¼–0.8 egg |
| Fish | 28–100 | 0.2–0.7 fish fillet |
| Legumes | 75–100 | 0.375–0.5 cup |
| Nuts | 50–75 | 1²/₃–2¹/₂ handful |
| Unsaturated fats | 20–80 | 5–20 teaspoons |
| Saturated fats | 11.8–27.7 | 3–7 teaspoons |
| Added sugar | 0 | 0 |

The linkages between heritage recipes, traditionally used food groups, and the average intake for healthy people and the planet are unclear.

Organic food production combines traditional and innovative food production methods with modern marketing principles [67]. In this context, the demand for organically grown foods has grown remarkably, both in developed and developing countries [68,69]. The perception and understanding of organic food production are based mainly on not using synthetic fertilizers and pesticides [70]. Moreover, organic agriculture can be characterized as less driven by off-farm inputs and better embedded in ecosystem functions; thus, it is likely a good model for productive and sustainable food production [68]. Although organic farming is a niche in most countries, it is on the verge of becoming mainstream in leading European countries [68]. Nevertheless, in less developed countries, it is often reported that the supply of organically grown food cannot fill the growing demand, the variety of organic food is somewhat limited, and the price of organic foods is much higher than non-organic foods [70–72].

In the face of such significant growth of organic practices in food production, there are also some risks related to a lack of knowledge. The food production practices that were until recently considered healthy and organic in the Global North are put in question. For example, in a recent article, the Environmental Working Group argues that the sewage sludge and biosolids that are commonly used in the US and Western Europe as organic fertilizer actually contain "forever chemicals" (known as PFAS) that infiltrate vegetable products [73]. Although human health effects from exposure to low environmental levels of PFAS are uncertain, the Centers for Disease Control and Prevention reported that " . . . studies of laboratory animals given large amounts of PFAS indicate that some PFAS may affect growth and development . . . may affect reproduction, thyroid function, the immune system, and injure the liver." [74]. This is still an area that needs a lot more work and research.

The understanding of which recipes are worthwhile to use in healthier diets is increasing and beneficial for the general health of the urban population; it often does not cause a lower vulnerability of the population to the impacts of climate change and does not lead to more organic urban farming or an increased growth and use of local crops.

### 4.2.4. Use of Local Ingredients

The attention, and the need, to use more local ingredients in the preparation of meals is increasing. The Farm to Fork concept [75] and the Slow Food Movement [76] illustrate this at the policy and objective level. On the ground, there are also many examples of restaurants that use local ingredients, such as in the UK [77], the US [78], or the Netherlands [79], where De Nieuwe Winkel experiments with botanical gastronomy picking ingredients themselves from their backyard, smelling and tasting the season and the region [80]. In the City of Monterrey, several chefs experiment with local recipes and locally grown ingredients, such as Chef Herrera, Fonda San Francisco [81], Chef Gonzalez, Pangea [82], or Chef Guajardo, El Vernaculo [83]. The need to use local ingredients and the eagerness of chefs to create dishes with these ingredients, however, is often not directly connected to the places where these ingredients are grown or the farmers that grow these crops.

The trend in many places and restaurants to make use of locally grown produce is increasing. This, however, is not always related to the establishment of urban farms in disadvantaged communities, support for the adaptation to climate impacts, or improvement in the intake of healthy ingredients.

### 4.2.5. Urban Agriculture

In the past decades, the interest in urban agriculture has grown [83–89]. The FAO has defined a framework for growing food in urban environments [90]. The focus lies on the production of food in urban and peri-urban areas [91]. Despite all this attention, urban agriculture projects are often limited to small-sized gardens or isolated food forests [92,93]. Examples of some of the most successful urban agriculture projects around the world are

the GrowUp Box in London [94], the Beacon Food Forest in Seattle [95], the Pasona Group office in Tokyo [96], The Farmery in North Carolina [97], Sky Greens, Lim Chu Kang area, Singapore [98], Brooklyn Grange, Brooklyn, New York [99], Deu Horta Na Telha in São Paulo [100], Prinzessinnengarten in Berlin [101], Urban Organics, St. Paul, Minnesota [102], and Lufa Farms in Montreal [103]. Some have permanently closed, others are in an isolated spot, or just very small. In Monterrey, a few urban agriculture projects, such as the Huerto Urbano [104], have been developed [105], with limited success.

There are significant differences in why people engage in food production in the Global North and Global South. For example, in the Global South, urban agriculture is mostly used to fulfill food security and nutritional needs. In comparison, urban agriculture in the Global North tends to be used for leading a more sustainable way of life or to create social ties within a community [106]. In a recent FAO (Food and Agriculture Organization of the United Nations) report on a project in the Democratic Republic of the Congo, it was mentioned that training peri-urban low-income city dwellers in developing urban agriculture significantly reduced food insecurity by doubling the vegetable intake of participants and reducing family's food expenditure by USD 20 to USD 60 [106]. Nevertheless, the authors warn that urban agriculture is still not proven safe in terms of health because food grown in urban areas could potentially be polluted from soils contaminated amidst busy roads and unclean water. In most developing countries, urban agriculture is not controlled or supported by governments; thus, there is no help in the provision of inputs such as seeds, fertilizers, chemicals, and technical advice. Ensuring access to clean water, land, and capacity building is crucial in supporting healthy food production in the global South [106].

The aim for growing nutritious and healthy food (Global South) and looking at urban food as a social and sustainable objective (Global North) could strengthen each other's purpose for a future sensitive food system. In this sense, it is advisable to share their perspectives so that the Global South could also aspire to emphasize the social ties within communities and a sustainable way of living and in turn, the Global North to start understanding to respond to urgent issues related to food security and nutritional food value. Moreover, the attention to urban agriculture worldwide may have different reasons in the Global North and South, it is becoming an important factor in urban development. However, this development is very often not integrated nor linked with social justice programs, the reduction in climate vulnerability, or connected to the use of crops in local restaurants or in traditional dishes.

### 4.3. Interconnected Components

For each of the research fields, valuable results are being achieved and published. However, this does not necessarily mean that the overall quality of the food consumed by urban residents is becoming healthier. The analysis makes clear that these research fields are only interconnected to a marginal level. Therefore, a series of 'missing links' are identified as follows:

- There is no link between the collection of traditional heritage recipes and the older population in disadvantaged communities;
- There is no connection between the potential workforce available in vulnerable communities and any training program for working in urban farming;
- There is no link between the local impacts of climate change and the positive effects of establishing a facility for growing organic food in the neighborhood;
- The growing of organic food in vulnerable communities (if at all) is not connected with the use of produce by chefs in higher-end restaurants;
- Cooking healthy dishes based on traditional recipes is disconnected from the consumption of healthy food in vulnerable communities.

These missing links stand in the way of improvements in each of the building blocks in which existing knowledge is generated. Moreover, the efforts in each of the sectoral scientific boxes do not lead to an improvement in healthy citizens, especially in vulnerable communities. To achieve this, an integrated approach is required in which the gaps are closed.

*4.4. Integration*

To overcome these 'missing links', it is proposed to create a virtuous circle in which the parts become interconnected and improve the intake of healthy diets/healthy people. The virtuous circle aims to constantly raise the quality of food and the resilience and health of citizens and improve daily living conditions. Once the first interventions that are closing the gaps take effect, they will start the development of increased impact that cannot be stopped. Several components are proposed to bridge the islands of knowledge.

4.4.1. Retrieve Traditional Recipes in (Vulnerable) Communities

In the disadvantaged community of la Campana, Monterrey, a long and intense conversation is established between the residents, policymakers, and researchers. This has led to mutual trust and a fertile ground for collaborative action. The older people living in the neighborhood have not (yet) been approached to share their knowledge about heritage food and traditional recipes. This conversation can, therefore, easily be ignited, and the outcomes may provide the content to be used for choices about the types of crops grown and the development of novel cuisine.

4.4.2. Menu and Diet

The traditional recipes form the input for novel dishes the restaurant chefs can develop. The chefs already work innovatively and retrieve their information from local produce, but not always the dishes driven by heritage food recipes. Moreover, the amounts of food types, given by the contextualized EAT-diet, are also used as input for the creation of the menu. When both the outcomes of the conversations with older residents and the EAT diet are used to inform and challenge the chefs, new dishes will be created, connecting the heritage, health, and growth of food.

4.4.3. Grow Organic Ingredients

The crops and produce that are required for running the restaurants based on heritage and healthy dishes need to be grown. Urban farms need to be realized in or near vulnerable communities to not only grow the produce but also allow residents to work and be trained for taking up a role in the urban farm. This strengthens the workforce and decreases social inequity. In addition to this, decreasing social inequity could be extended to raising the bar in the future food cycle significantly higher. This could also include the education of younger members of these communities to pursue skills and employment in pathways all along the food cycle, trained to cook as a chef, restaurateur, or food tradesman. This would support the long-term uptake of young people in all sections of the food cycle.

In this research project, the first urban farm is projected to be designed in Los Pinos, a public space in the la Campana neighborhood. The farm will be managed by the community members and grow the produce that is demanded in an organic way, e.g., without the use of chemical fertilizers, pesticides, and herbicides. To prevent waste, such as microplastics, from entering the food system [107], the use of sewage sludge is prevented. Growing food is a seasonal activity; hence, it must be programmed and planned according to the climate and, at the same time, meet the demand of the restaurants. To overcome a possible mismatch in produce and demand, an old warehouse will be rented to operate as a storage hub for all organically grown food. The restaurant chefs can obtain the required ingredients when they need them in this centrally located warehouse.

4.4.4. Prepare/Cook; Innovate Traditions

The locally grown organic ingredients will then directly be used by the chefs that design the dishes and menus. They can obtain the ingredients from the storage and use them to cook daily in their restaurants. This way, a close connection is established between the type of food grown and the type of dishes cooked.

### 4.4.5. Feed the Food Back to the Community

The final closing element is to feed the healthy dishes back to the community that came up with the traditional recipes and has grown the produce themselves. To provide a facility where the residents can eat their healthy dishes, the restaurants will open a pop-up restaurant in the community, where local residents are trained to cook, and the community is invited to eat cheap and healthy.

By integrating these components in between and within the more thematic scientific research fields, they start to act as connectors, multiplying the outcomes of specific research. They all link fundamental and more abstract knowledge with applications of that knowledge in practice. This enhances the impact of academic findings and improves the lives of residents in vulnerable communities. The local food circle can only be virtuous when these gaps are closed.

## 5. Conclusions

In this article, a novel model is developed which emphasizes the essential components that need to be implemented in closing the food circle. Many fields of science are isolated and deliver fragmented results. This leaves the academic outcomes behind without impact on society.

The isolated fields of climate change research and policy, collection of heritage recipes, urban agriculture projects, social equity programs, and the use of local ingredients in the kitchen, deliver results for their own section of reality, without transforming societies. This has profound consequences for the mental and physical health of citizens, especially for those already living in poor conditions.

By adding components such as retrieving recipes in the community, realizing an urban farm where people from the community are trained and work, and feeding the healthy recipes back into these disadvantaged neighborhoods, a multiplicity of objectives is achieved as follows:

- The neighborhood is greened and becomes more resilient to climatic impacts;
- Social inequity is reduced;
- Heritage food is preserved;
- People eat a healthier diet that helps them to stay within planetary boundaries;
- Chefs apply locally grown organic produce, stay innovative and feed healthier food to their customers.

The circle is virtuous because when this development is ignited via implanting the missing links, the progress will be unstoppable, and using healthy ingredients, locally grown produce, and novel dishes will continue increasing the number of healthier people.

There is a lot of support for and awareness of the urgency to eat healthier, leading to potentially much lower costs for health care. There is a willingness in the vulnerable communities, and the local chefs are eager to start using more organic and local ingredients in their traditionally driven menus, whilst adapting to current times and tastes. The policies are in place to support the concept; however, the proof of the pudding lies in eating, e.g., the implementation and execution of the missing links. This trajectory of transformation is sometimes hindered by the vested interests of companies that produce industrialized processed foods, and this makes it difficult to change course. There is a risk that the food system remains fragmented due to the unclarity of responsibilities for the entire loop and who is responsible and willing to invest in closing the identified gaps.

**Author Contributions:** Conceptualization, R.R. and A.E.M.; methodology, R.R.; validation, R.R., A.E.M. and A.K.; formal analysis, R.R. and A.K.; investigation, R.R., A.E.M. and A.K.; resources, R.R.; data curation, R.R. and A.K.; writing—original draft preparation, R.R.; writing—review and editing, A.K.; visualization, R.R.; supervision, R.R.; project administration, R.R.; funding acquisition, R.R. All authors have read and agreed to the published version of the manuscript.

**Funding:** This research received no external funding.

**Institutional Review Board Statement:** Not applicable.

**Informed Consent Statement:** Not applicable.

**Conflicts of Interest:** The authors declare no conflict of interest.

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
