# Peer review of "Creating a Virtuous Food Cycle in Monterrey, Mexico"

_sustainability, doi:10.3390/su15107858_

Round 1

Reviewer 1 Report

Table 1 is not clear. there should be a clear indication about diet type and its to metal health aspect. 

Title need modification as it does not match the contents of manuscript.

Methodology should be described in detail for reproduction of the model and research in any part of the world. data collection and interpretation is vague and must be improved. 

selection of diet and relation of mental health of the region must be part of the article. 

Language is acceptable with minor improvement especially in methodology part. 

Reviewer 2 Report

Line 35: “reduces people’s physical and mental health” actually not all foods reduces health, only those processed and contains synthetic additives such as synthetic antioxidants or nitrite in sausages or under inappropriate storage and handling they causes threat to health, either by the additives or through the formation of lipid and protein radicals resulting from oxidation by ROS. Therefore, line 35 is too general statement about the current food system, and need to be rewritten appropriately.

Line 36: “traditional recipes and ingredients” sometimes farmers and villagers uses much more oils and salt when prepare foods, and this may increase the risk of health. However, I agree that traditional recipes usually use fresh ingredients such as vegetables but if still uses high salt and oils at home it can increase health risk. I suggest rewrite the sentence appropriately.

Line 61: “overconsumption” in this case authors should provide the consumption rate of such processed foods in the city of Monterey (kg/person/year) and if it is too high compare it with global statistic or in the Mexico of consuming such foods, furthermore is there any evidence that people in Monterey have higher rate of diabetes or obese people compared to other cities in the country?

Line 64: “Especially children are increasingly obese and overweight” should include a reference from health authorities of Mexico. The reference provided (6) (Nicklas et al., 2001) is too old and cannot be reliable for the 2023 year.  

Lines 145-146: “Without awareness of the positive health implications of traditional food patterns” authors still did not mention what are the main drivers for the better health effects of traditional foods mainly prepared in villages or by grandmothers?

I think the whole manuscript is too general without the exact scientific data or rational evidence on the items are being discussed. It needs more scientific information especially from health ministry or the relevant regulatory authorities.

Can authors add some types of traditional foods that are being considered healthier compared to processed foods in Monterey?

Reviewer 3 Report

To highlight heritage recipes as a solution for closing healthy food cycle gaps is a significant game changer for disadvantaged communities. This work should begin a wider conversation on how those traditions can significantly influence urban, programmatic, educational and aspirational advancement.

A minor adaptation. The following sentence is confusing. 'Negatively related to perceived stress'  reads to me as if healthy foods are the reason for increased stress and depression.

See: On the contrary, the consumption of healthy food, such as fresh fruits, salads, and cooked vegetables, is negatively related with perceived stress and depressive symptoms [17].

A minor spelling adaptation. Limit - limiting

In an integrated way this could be achieved by simultaneously limiting agricultural expansion, for instance by more localized and nutritious foodscapes, reduce emissions, and expand forested lands.

Suggestions:

1.     Would be highly beneficial to graphically show at least one example of a food heritage traditional recipe, its ingredients, preparation, the community added value (inclusivity/cohesion), etc… Otherwise the study appears socially detached from the history and social benefit of how communities can reveal their own source of sustainable growth and rejuvenation. This is after all the power of the study.

2.     The building blocks/case studies (in which there are gaps in-between) are described in a scrolling format with no apparent narrative, this isn’t so important, however, a more coherent summary of each would benefit the overall readability of the work. Case studies (for example, urban agriculture) are listed in paragraphs and in series, perhaps for impact select one and elaborate further with an addition of illustrations/diagrams etc…

3.     The dietary information on ingredients is a data monologue, I understand its inclusion, however, I would suggest a sharper summary of what that data represents.

The global difference in North and South is a key point. However, this circumstance somehow appears to end as a statement, no comment on how to move forward within the frame of this study is offered:

For example, in the Global South urban agriculture is mostly used to fulfill food security and nutritional needs. In comparison, urban agriculture in the Global North tends to be used for leading a more sustainable way of life or to create social ties within a community [100].

Could it not be proposed that (using this study as a lens) that the challenges and aspirations of both hemispheres, North / South, should be shared, more balanced, and interpreted in some way. This arguably is one of the objectives of this work, that the South aspires to a sustainable way of life / to create social ties within a community, and vice versa, that the North should begin to understand and respond more urgently to food security and nutritional needs.

The statement below can be argued, however could not/should not the younger members of disadvantaged communities be supported educationally/financially in pursuits of higher aspiration. Could they not also find paths all the way along the ‘food cycle’ such as training to be Chefs, the restaurateur, the trader? Here I suggest ‘trained to cook’ needs a far greater reach and longer-term ambition. It was argued that disadvantaged communities, those in poverty, are disadvantaged due to a lack of education, can that educational bar be raised significantly higher in this food cycle future? Could/should your argument for decreasing social inequity extend to this?

Urban farms need to be realized in or near the vulnerable communities to not only grow the produce, but also allow residents to work and be trained for taking up a role in the urban farm. This strengthens the workforce and decreases social inequity.

No comments additional to those above.

Round 2

Reviewer 2 Report

Thanks for adequately responding to queries